# RadBone: bone toxicity following pelvic radiotherapy – a prospective randomised controlled feasibility study evaluating a musculoskeletal health package in women with gynaecological cancers undergoing pelvic radiotherapy

Victoria Chatzimavridou Grigoriadou [1,2] Lisa H Barraclough,[3]
Ivona Baricevic-Jones,[4] Robert G Bristow,[5] Martin Eden,[6] Kate Haslett,[3]
Karen Johnson,[3] Rohit Kochhar,[7] Zoe Merchant,[8] John Moore,[9,10]
Sarah O'Connell,[7] Sally Taylor,[11] Thomas Westwood,[7] Anthony David Whetton,[12,13]
Janelle Yorke,[3,14] Claire E Higham[1,2]

**Correspondence to**
Dr Claire E Higham;
claire.higham2@nhs.net

## ABSTRACT

**Introduction** Patients receiving radiotherapy are at risk of developing radiotherapy-related insufficiency fractures, which are associated with increased morbidity and pose a significant burden to patients' quality of life and to the health system. Therefore, effective preventive techniques are urgently required. The RadBone randomised controlled trial (RCT) aims to determine the feasibility and acceptability of a musculoskeletal health package (MHP) intervention in women undergoing pelvic radiotherapy for gynaecological malignancies and to preliminary explore clinical effectiveness of the intervention.

**Methods and analysis** The RadBone RCT will evaluate the addition to standard care of an MHP consisting of a physical assessment of the musculoskeletal health, a 3-month prehabilitation personalised exercise package, as well as an evaluation of the fracture risk and if required the prescription of appropriate bone treatment including calcium, vitamin D and—for high-risk individuals— bisphosphonates. Forty participants will be randomised in each group (MHP or observation) and will be followed for 18 months. The primary outcome of this RCT will be feasibility, including the eligibility, screening and recruitment rate, intervention fidelity and attrition rates; acceptability and health economics. Clinical effectiveness and bone turnover markers will be evaluated as secondary outcomes.

**Ethics and dissemination** This study has been approved by the Greater Manchester East Research Ethics Committee (Reference: 20/NW/0410, November 2020). The results will be published in peer-reviewed journals, will be presented in national and international conferences and will be communicated to relevant stakeholders. Moreover, a plain English report will be shared with the study participants, patients' organisations and media.

**Trial registration number** NCT04555317.

## STRENGTHS AND LIMITATIONS OF THIS STUDY

⇒ The RadBone is the first randomised controlled trial to assess a musculoskeletal health package aimed to prevent radiotherapy-related insufficiency fractures.

⇒ A feasibility economic evaluation will allow future assessment of this complex intervention's cost-effectiveness.

⇒ Planned longitudinal proteomic analyses may reveal mechanistic insights and promising treatment targets.

⇒ A prospectively published detailed protocol increases the transparency and allows for peer review of the methodology used.

⇒ This study is not blinded and lacks an active comparator, hence, it is susceptible to performance and detection bias.

## INTRODUCTION

In 2015, there were 2.5 million people in the UK with a diagnosis of cancer and this number is expected to rise to 4 million by 2030.[1] As a result of the continuing improvement in early detection of disease and improved treatment efficacy, a significant proportion are living long beyond their cancer diagnosis. However, estimates suggest that currently over 500 000 people living with and beyond cancer have one or more physical or psychosocial consequences of their cancer or its treatment that affect their lives on a long-term basis. These consequences also have a substantial implication in terms of National Health Service (NHS) resources.

Patients receiving radiotherapy are at risk of developing radiotherapy-related bone toxicity, in particular radiotherapy-related insufficiency fractures (RRIFs). Incidence of RRIFs following pelvic radiotherapy has been reported between 1.7% and 89% and occurring between 3 and 20 months post-radiotherapy. The wide variation in reported incidence depends on imaging modality and radiological reporting standards, symptomatic versus asymptomatic fractures, radiotherapy dose and underlying tumour type.[2] A recent meta-analysis of over 400 patients with RRIFs following pelvic radiotherapy for gynaecological cancers suggested an overall incidence of 14%.[3] Over 30 studies have been published since the 1990s describing >1000 patients with pelvic RRIFs. This literature is notable for being almost exclusively retrospective in nature, a sparsity of baseline assessment of bone density and fracture risk, the absence of patient-reported outcome measures (PROMs) used to assess quality of life (QOL) and no primary preventative or secondary management intervention studies.[4 5]

The devastating effects of osteoporotic fragility fractures on morbidity and mortality and the economic cost are well described.[6] Pelvic insufficiency fractures may also increase mortality[7] but these data reflect an elderly population with multiple comorbidities and the applicability to the pelvic radiotherapy population is not well defined. In addition, there are no pelvic RRIF studies reporting QOL as an outcome measure. However, the anxiety, pain, reduced mobility and increased morbidity associated with these have been described, with a number of patients requiring hospital admission for assessment and pain control.[8] Therefore, formal studies of QOL and PROMs are much needed, considering the wide range of pelvic radiotherapy toxicities.[9]

While a small number of studies, confirmed in a recent meta-analysis,[3] suggest osteoporosis as a risk factor in pelvic RRIFs, unlike the strong evidence base for bisphosphonate use in primary and secondary prevention of fragility fractures, there is no such evidence for RRIFs.[5] A small non-controlled study demonstrated intravenous zolendronic acid administration prior to spinal radiotherapy led to a lower prevalence of radiotherapy bone toxicity than expected[10] and a single randomised prospective study in patients undergoing spinal radiotherapy for metastatic disease demonstrated that intravenous zolendronic acid reduced urinary markers of collagen cross-linking.[11]

Contradictory data from animal studies around the protective effects of bisphosphonates on RRIFs limit our understanding of the pathophysiology and therapeutics of RRIFs. Animal studies using whole mouse radiation have demonstrated an early activation of bone resorption in the 5 days following low dose (2 Gy) of radiotherapy which was reduced by subcutaneous administration of risedronate immediately following irradiation.[12] In contrast, a focal radiation technique in mice (using a small animal radiation research platform), arguably a more physiological representative method of irradiation,

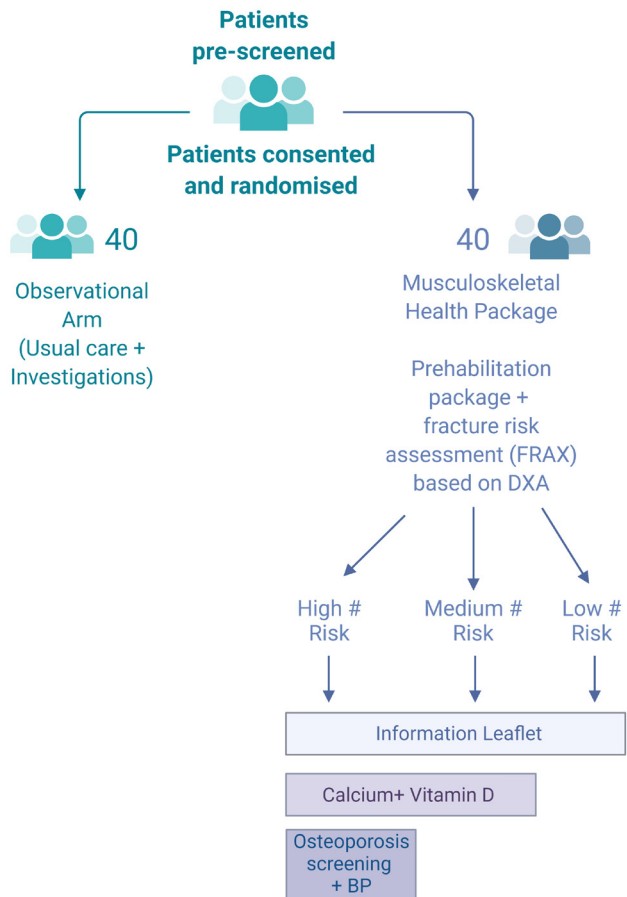

**Figure 1** Recruitment, randomisation process and description of the stratified interventions (#: fracture). BP, blood pressure; DXA, dual-energy X-ray absorptiometry; FRAX, fracture risk assessment.

demonstrated that alendronic acid did not prevent the radiation-induced trabecular bone loss but that this was prevented by blocking osteoblast apoptosis with PTH 1–34.[13]

The RadBone is the first open-label prospective randomised controlled trial (RCT) to determine the feasibility and acceptability of a musculoskeletal health package (MHP) intervention in women undergoing pelvic radiotherapy for gynaecological malignancies and inform power calculations for a definitive RCT. Moreover, this feasibility trial will also explore potential implications on the incidence of RRIFs, QOL and other clinical effectiveness and safety outcomes, as well as providing indicative estimates of the intervention's cost-effectiveness.

## METHODS AND ANALYSIS
### Study design
See figure 1.

### Study setting
The planned study is a prospective randomised controlled feasibility trial of 80 patients with gynaecological

malignancy (cervical and endometrial) undergoing pelvic radiotherapy at the Christie Hospital NHS Foundation Trust in Manchester, UK (a tertiary referral oncology centre). The study opened for recruitment in May 2021, and the estimated primary completion date is in November 2022 and study completion date in June 2023.

### Eligibility criteria

Individuals aged over 18 years, with a histologically confirmed endometrial or cervical cancer undergoing potentially curative or adjuvant radiotherapy will be eligible, provided they are able and willing to provide an informed consent to participate.

The exclusion criteria are: (i) age <18 years or >85 years; (ii) pre-existing bone conditions such as osteoporosis treated with bisphosphonates in the previous 5 years, fibrous dysplasia, osteogenesis imperfecta or other metabolic bone conditions; (iii) home address outside Greater Manchester; (iv) contraindication or intolerance of magnetic resonance scanning.

### Interventions

Women undergoing radiotherapy for a gynaecological malignancy will be randomised to an observation group and will receive standard assessment and care, following the current local clinical pathway, or an intervention group that will receive an MHP, in addition to standard assessment and care and will be followed for 18 months.

Patients randomised to the MHP arm will receive (i) a physical assessment of musculoskeletal health and a 3-month prehabilitation personalised exercise package as part of the Greater Manchester Prehab4Cancer programme,[14] (ii) a fracture risk assessment (FRAX) based on baseline dual-energy X-ray absorptiometry (DXA) bone mineral density (BMD) and (iii) treatment for bone health according to national UK recommendations, that is, standard of care for prevention of fragility fractures, by being subdivided into three groups (low risk, medium risk and high risk).

Patients with a normal BMD and a FRAX score below the National Osteoporosis Guideline Group (NOGG) recommended treatment line will be considered low risk. Medium risk is defined as osteopenia on the DXA, with FRAX score below the NOGG treatment line. Finally, those with osteopenia and a previous vertebral or hip fracture, or a FRAX score above the NOGG recommended treatment line will be considered high risk.

Low-risk patients will be provided with a copy of the Royal Osteoporosis Society 'healthy living for strong bones' leaflet. In addition to the leaflet, medium-risk patients will receive calcium (1000 mg once daily) and vitamin D (800 IU/day) supplementation. The same interventions will be offered to high-risk patients, who will also undergo secondary osteoporosis screening (blood tests) and will receive oral alendronate 70 mg once weekly, in the absence of contraindications. Annual intravenous zolendronic acid infusion will be considered as an alternative where appropriate.

Those randomised to the observation arm will remain blinded to the results of the evaluations until the end of the study unless a fragility fracture or RRIF develops during the study.

### Prehabilitation exercise programme (Prehab4Cancer)

All patients randomised to the MHP arm of the study will be offered a bespoke prehabilitation exercise programme via the Prehab4Cancer programme in Greater Manchester. The MHP arm patients will be referred to the Prehab4Cancer team via electronic referral immediately following randomisation. Allocated patients will be individually assessed by the Prehab4Cancer team according to their usual protocols and assigned an appropriate prehabilitation programme. Duration of the programme is 12 weeks from the first assessment and participation will be encouraged, as tolerated. The Prehab4Cancer and recovery programme is community-based, which incorporates exercise (cardiovascular and muscle strengthening/resistance training), nutritional screening and advice and well-being support. Further details of programmes' assessment tools and the stratification of interventions are described by Moore et al[14] and can be found in www.prehab4cancer.co.uk. The current scope of this protocol is to evaluate feasibility of participants' engagement in this face-to-face and remote prehabilitation service both pretreatment and during treatment.

### Baseline and follow-up evaluation

As described in figure 2, baseline evaluations will include a bone health assessment with DXA BMD measurement and completion of a bone health questionnaire. PROMs

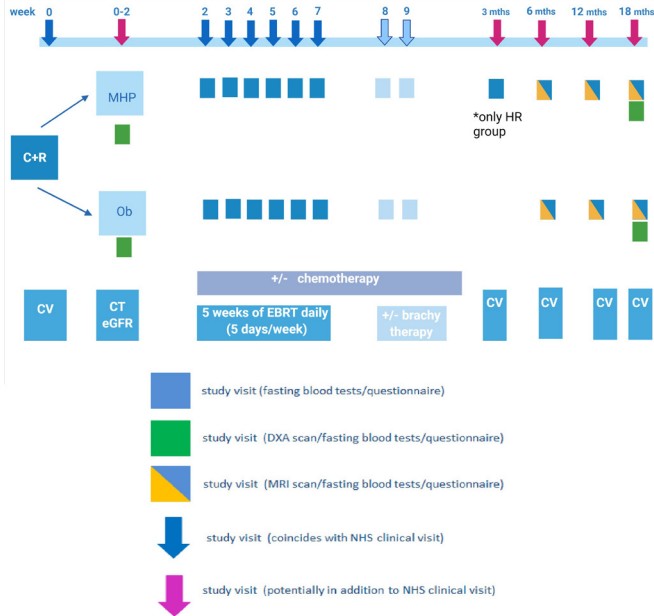

**Figure 2** Study flow chart; assessments and outcome time-points. C+R, consent and randomisation; CV, clinic visit; DXA, dual energy X-ray absorptiometry; EBRT, external beam radiotherapy; eGFR, estimated glomerular filtration rate; HR, high risk; MHP, musculoskeletal health package arm; NHS, National Health System; Ob, observational arm.

will also be captured. Finally, fasting serum and plasma blood samples will also be collected.

At 6, 12 and 18 months postradiotherapy, all patients will undergo a pelvic MRI assessment for RRIFs, PROMs assessment and fasting blood sampling. During the final visit, at 18 months, all patients will have a DXA BMD scan and physical assessment of their musculoskeletal health. If signs or symptoms compatible with an RRIF are described outside the study visits study participants will be assessed and managed following the current clinical pathways.

### Imaging studies

DXA scans of the total hip, femoral neck, L1–L4 spine and Trabecular Bone Score (TBS) assessments will be performed on a single DXA scanner (Hologic Horizon A SN 300792M V.5.6.07 with TBS V.3.0.2 calibrated to the above scanner) at the Christie NHS Foundation Trust as per local protocol. These will be undertaken by two technicians trained in conducting DXA. Images will be reviewed, validated and interpreted by the lead investigator (CEH). The femoral neck BMD ($g/cm^2$) will be used in conjunction with a standardised DXA questionnaire to complete FRAX calculation.

Pelvic MRI scans will be performed at 6, 12 and 18 months on a 1.5 T MRI scanner at the Christie Hospital by trained radiographers in accordance with the study imaging protocol. Four pelvic sequences will be performed per patient (5 mm slice thickness, field of view 400 mm; coronal T1, coronal Short Tau Inversion Recovery (STIR) (inversion time 150 ms), axial T1 and axial STIR (inversion time 165 ms). These will correspond to routine follow-up scans where possible. All bone sequence scans will be dual reported by two consultant radiologists who will document the presence of fracture and their confidence in its presence, fracture location, fracture line, bone marrow oedema and other abnormalities.

### Biochemical studies

Fasting blood tests will be performed at baseline, weekly during radiotherapy (visits 2–10, 1 day prior to chemotherapy if receiving) and at 6, 12 and 18 months in all patients. Patients allocated to the MHP high-risk arm and started on oral bisphosphonate therapy will have an additional bone turnover marker blood test at 3 months to assess bisphosphonate efficacy. All samples will be taken simultaneously with routinely collected clinical blood samples where appropriate.

As part of the MHP intervention arm, blood will be sampled, analysed and assessed at baseline, 6, 12 and 18 months for the measurement of full blood count, urea and electrolytes, liver function tests, parathyroid hormone, vitamin D, thyroid function test, oestradiol, haemoglobin A1c, procollagen type 1 amino-terminal propeptide (P1NP) and the beta-C-terminal telopeptide (CTX). Moreover, in the observation arm, serum samples will be collected, processed and stored at −80°C for batch analysis at the end of the study.

Additional fasting blood samples will be collected at all timepoints mentioned for longitudinal analysis of bone turnover markers and for proteomic analysis. These samples will be processed and stored at −80°C, following local standard operating procedures (SOPs), for batch analysis at the end of the study. Bone turnover markers will be evaluated using ELISA techniques and will include CTX, N-terminal telopeptide (NTX), P1NP, osteocalcin, Tartrate-resistant acid phosphatase 5b (TRAcP5b) and bone alkaline phosphatase (ALP).

### Sequential Window Acquisition of all Theoretical Mass Spectra

Proteomic analysis will be conducted at the Stoller Biomarker Discovery Centre, following local SOPs.[15] Samples will be analysed by a data independent acquisition method known as Sequential Window Acquisition of all Theoretical Mass Spectra (SWATH-MS) with a microflow liquid chromatography-mass spectrometry system comprising an Eksigent nanoLC 400 autosampler and an Eksigent nanoLC pump coupled to a SCIEX 6600 Triple-TOF mass spectrometer (68 min run-time). When SWATH maps are generated, the presence and abundance of plasma proteins will be quantified using published plasma reference libraries. Differential expression analysis will be used to identify candidate biomarkers using artificial intelligence approaches. Linear regression will be used to detect correlations with the presence of RRIFs and BMD.

Few longitudinal studies have tracked proteins of interest over the whole course of radiotherapy from pretreatment baseline through to follow-up. We have undertaken one pilot that shows the potential value of this work.[16] Other studies that have investigated this have demonstrated distinguishing profiles with groups of approximately n=30. Two preradiotherapy baseline samples will be used to assess natural variation and comparison with the variance of measurements following radiotherapy and further comparison between the MHP and observation arm (n=40 per group).

Electronic data will be pseudoanonymised (coded) to protect the identity of the participants.

### PROMs and health utilisation proforma

PROMs will be collected either as electronic PROMs (using the myChristie, myHealth application) or paper-based PROMs at baseline 6, 12 and 18 months.

The evaluated PROMs will include the adapted pelvic patient-reported outcome version of the common terminology criteria for adverse events (PRO-CTCAE) assessment, the Short Musculoskeletal Function Assessment (SMFA) modified for lower limb, the 5-level version of the EuroQol tool (EQ-5D-5L) and a tailored health utilisation proforma.

The CTCAE pelvic questionnaire will include as measures bowel questions scored out of 22, urinary questions out of 19 and sexual questions out of 8, with a total out of 49; a higher score indicates worse QOL. The adapted SMFA questionnaire includes 39 questions, with

a minimum possible score of 39 and maximum of 195; scores are standardised with high scores indicating poor function.

## Criteria for discontinuing

Participants may decide to withdraw from the study at any time. Discontinuation of the study participants may occur as a result of investigator decision, safety concerns and significant non-compliance to the protocol or incorrect enrolment. Reasons for discontinuation will be captured.

As this is a feasibility study, participants may decide to discontinue their participation in certain aspects of the study (eg, declining the prehabilitation programme or deciding not to take recommended medications). The participants can continue with the study and the details will be captured in the case report form.

## Outcomes

The primary outcomes for this feasibility study will inform the design and power calculations for a definitive UK multicentre RCT. These are:

1. Eligibility and screening rate: proportion of patients eligible for the study from patient population (assessed at baseline).
2. Recruitment and study group allocation rate: number and proportion of eligible patients recruited, randomised and allocated to appropriate study populations (assessed 2 weeks postconsent).
3. Intervention fidelity rate: number and proportion of patients completing the elements of the study (assessment visits, prehab exercise programme, prescribed medications, QOL questionnaire) (assessed at the end of study, at 18 months).
4. Attrition rate: number of patients lost to follow-up (assessed at the end of study, at 18 months).
5. Patient and physician acceptability assessed with electronic questionnaires (change from baseline assessed at 6, 12 and 18 months).
6. Health economic analysis: within-trial cost-effectiveness analysis to demonstrate feasibility of health economic data collection and analysis in a multicentre RCT (change from baseline assessed at 6, 12 and 18 months).

The secondary outcomes are:
1. Incidence of pelvic RRIF (assessed at 6, 12 and 18 months postradiotherapy).
2. Longitudinal change in BMD and fracture risk using FRAX (assessed at baseline and 18 months).
3. Longitudinal change in biochemical markers of bone turnover (change from baseline assessed at 2, 3, 4, 5, 6, 7, 8 and 9 weeks and at 6, 12 and 18 months).
4. QOL assessment: adapted CTCAE pelvic questionnaire and SMFA adapted to lower limbs (change from baseline assessed at 6, 12 and 18 months).

Exploratory end points include identification of predictive markers of RRIFs (radiomic, proteomic, BMD) and exploratory measurement of proteomic biomarkers of bone turnover during pelvic radiotherapy.

## Sample size

No formal power calculation has been performed as this is a feasibility study. The study will collect initial data such as measures of location and variability for key outcome measures. It is recognised that in general, 30 patients are required in order to estimate such parameters.[17] For this study, a total of 80 patients will be recruited and randomised with equal probability to either the MHP or observation arms (ie, 40 per group). Assuming attrition rates of 15% per group, at least 30 should remain in each arm. This should be sufficient to assess the feasibility of a larger RCT study and estimate group means, SD and percentages for key outcomes.

## Recruitment

Eighty patients will be recruited over an 18-month period, approximately 4 patients per month. As this is a feasibility study, there will be no interim analysis of study results.

## Assignment of interventions

Consenting, eligible participants will be randomised to the MHP or observation group using a validated online service; sealedenvelope (https://www.sealedenvelope.com). A permuted block (block size: 4) randomisation protocol will be used with a 1:1 allocation (MHP to observation arm).

## Data collection, management and analysis
### Statistical and health economic analysis

As this is a feasibility study, it will not involve hypothesis testing to identify whether the intervention has had an impact. Instead, data analysis will be descriptive, focusing on the percentage of patients in each group developing RRIFs and risk factors for this. Means and a measure of variation will be calculated for each secondary outcome. These data, along with estimates of recruitment and attrition rates, will help inform a power calculation for the definitive trial.

A within-trial cost-effectiveness analysis[18] will be undertaken from the perspective of the UK NHS. Cost data for the intervention arm will reflect resource use associated with the MHP and treatment costs for both the control and intervention arm will be taken into account. Resource use will be extracted from patient records and the healthcare utilisation proforma. Relevant sources (eg, NHS reference costs) will be used to identify unit costs. Health-related QOL scores will be generated using the EQ-5D-5L at baseline and at each of the three follow-up time points (6, 12, 18 months).

A descriptive analysis of the costs and outcomes data will be completed focusing on: (a) whether the EQ-5D-5L and SMFA are able to adequately capture differences in health status before and after implementation of the MHP and across both treatment arms of the study; (b) whether the resource-use survey is able to record data necessary to enable a full cost-effectiveness analysis; (c) the nature of missing data for the EQ-5D-5L, SMFA and resource-use survey to assess responses, sensitivity and any patterns within the missing data.

A within-trial cost-effectiveness analysis will be conducted to provide an indicative estimate of cost-effectiveness. Between-arm differences in costs and outcomes will be expressed as an incremental cost-effectiveness ratio (ICERs): the cost per quality-adjusted life year gained from the intervention compared with usual care. ICERs will also be calculated using the SMFA in an additional scenario analysis.

## Trial oversight

An internal trial management group will be convened for the study, consisting of the chief investigator, project manager, Clinical Trials administrator, research nurse and a representative of the research and innovation division as core members. The group will meet monthly. The study sponsor (Christie Hospital NHS Foundation Trust) will monitor the conduct of the trial.

**Author affiliations**
[1]Department of Endocrinology, The Christie Hospital NHS Foundation Trust, Manchester, UK
[2]Manchester Academic Health Science Centre, The University of Manchester, Manchester, UK
[3]Department of Clinical Oncology, The Christie Hospital NHS Foundation Trust, Manchester, UK
[4]Stoller Biomarker Discovery Centre, Division of Cancer Sciences, School of Medical Sciences, Faculty of Biology, Medicine and Health, University of Manchester, Manchester Academic Health Science Centre, Manchester, UK
[5]Division of Cancer Sciences, The University of Manchester, Manchester, UK
[6]Manchester Centre for Health Economics, The University of Manchester, Manchester, UK
[7]Department of Radiology, The Christie NHS Foundation Trust, Manchester, UK
[8]Greater Manchester 'Prehab4Cancer and Recovery programme'/Highly Specialist Occupational Therapist, GM Cancer alliance hosted by the Christie NHS Foundation Trust, Manchester, UK
[9]GM Cancer Clinical Director for Prehabilitation and Recovery, University of Manchester and Manchester Metropolitan University, Manchester, UK
[10]Anaesthetics and Intensive Care Medicine, Manchester University NHS Foundation Trust, Manchester, UK
[11]The Christie Patient Centred Research Team, The Christie School of Oncology, The Christie NHS Foundation Trust, Manchester, UK
[12]Stoller Biomarker Discovery Centre, The University of Manchester, Manchester, UK
[13]Division of Cancer Sciences, Faculty of Biology, Medicine and Health, The University of Manchester, Manchester, UK
[14]Christie Patient-Centred Research, Division of Nursing, Midwifery & Social Work, The University of Manchester School of Health Sciences, Manchester, UK

**Contributors** CEH developed the protocol, MRC application and ethics application for the study. RGB is the MRC-CARP academic partner to CEH and contributed to the study design and protocol. KJ, LHB, KH contributed to the development of the gynae-oncology aspects. RK, SO'C, TW contributed to the development of the MRI radiology aspects. ZM, JM contributed to the development of the Prehab4Cancer aspects. ST, JY contributed to the PROMs development. ME contributed to the Health Economic Analysis. ADW, IB-J contributed to the proteomic and data analysis. VCG and CEH prepared the manuscript. All authors: critically reviewed and commented on the manuscript.

**Funding** This work is partially supported by an MRC-NIHR Clinical Academic Research Partnership award: grant number MR/T024887/1. Funding is being sought for Health Economic Analysis, QOL and PROMs development and longitudinal proteomic analysis. Equipment used in the Stoller Biomarker Discovery Centre is funded by a donation received from the Stoller Charitable Trust and a research grant awarded by the Medical Research Council (MR/M008959/1). MRC/EPSRC Molecular Pathology Node provided additional financial support by a grant from the Medical Research Council and Engineering & Physical Sciences Research Council (MR/N00583X/1).

**Competing interests** None declared.

**Patient and public involvement** This protocol was developed with the participation of the Christie pelvic radiotherapy user group and supported by the Pelvic Radiation Disease Association (PRDA).

**Patient consent for publication** Not applicable.

**Provenance and peer review** Not commissioned; externally peer reviewed.

**ORCID iD**
Victoria Chatzimavridou Grigoriadou http://orcid.org/0000-0002-9064-0532

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
