## [Reviewer comments · BMJ Open]

ARTICLE DETAILS

TITLE (PROVISIONAL)	RadBone: Bone Toxicity following Pelvic Radiotherapy. A prospective randomised controlled feasibility study evaluating a musculoskeletal health package in women with gynaecological cancers undergoing pelvic radiotherapy.
AUTHORS	Chatzimavridou Grigoriadou, Victoria; Barraclough, Lisa H; Baricevic-Jones, Ivona; Bristow, Robert; Eden, Martin; Haslett, Kate; Johnson, Karen; Kochhar, Rohit; Merchant, Zoe; Moore, John; O'Connell, Sarah; Taylor, Sally; Westwood, Thomas; Whetton, Anthony; Yorke, Janelle; Higham, Claire E

VERSION 1 – REVIEW

REVIEWER	Faithfull, Sara University of Surrey, School of Health Sciences Chair NCRI LWBC Late conséquences research sub committee
REVIEW RETURNED	10-Nov-2021

GENERAL COMMENTS	Thank you for such an interesting protocol. The topic is a much needed area of research and intervention. There are ta couple of minor changes that would help with clarity. 1) The study design figure 1. shows process of recruitment and randomisation but would benefit from including the assessment and outcome time points.2) The introduction line 42-49 this paragraph seems out of place with a discussion of quality of life which is then followed by mouse biomarkers etc. This paragraph could be summarised and added to the paragraph above.3) Figure 2 highlights the interventions but is clouded by the assessments and outcomes that will be conducted. This would be easier to understand if just focused on the stratified intervention.4) The MHP includes personalised physiotherapy and exercise, it will be critical to record how this is personalised for future replication, this is not described and may vary considerably. This may need documenting re FITT ie frequency, intensity, type and timing so that this can be analysed. This maybe already documented in the current prehabilitation programme but this need clarifying in the protocol as adherence and intensity are important for intervention fidelity and different clinicians can personalise quite differently ie are their core exercises or resistance training and what is adapted. This was a very interesting paper to read and look forward too eeing the outcomes of the study.
--

VERSION 1 – AUTHOR RESPONSE

Reviewer: 1

Prof. Sara Faithfull, University of Surrey

Comments to the Author:

Thank you for such an interesting protocol. The topic is a much needed area of research and intervention. There are a couple of minor changes that would help with clarity.

Comment 1: The study design figure 1. shows process of recruitment and randomisation but would benefit from including the assessment and outcome time points.

Comment 3: Figure 2 highlights the interventions but is clouded by the assessments and outcomes that will be conducted. This would be easier to understand if just focused on the stratified intervention.

Thank you for your constructive comments. We note that figures 1 and 2 were not adequately focused and hence we have updated figure 1 to show the recruitment and randomisation process and make clearer the stratified intervention in the different groups. We did not include the latter in figure 2, as this figure is placed in the section of "Baseline and Follow-up Evaluation". Instead, we have simplified it, to focus on the assessment and outcome time points. (Legends: Lines 458-465, figures not embedded to main document as per Editorial Office instructions)

Comment 2: The introduction line 42-49 this paragraph seems out of place with a discussion of quality of life which is then followed by mouse biomarkers etc. This paragraph could be summarised and added to the paragraph above.

Thank you for your comment. We have accordingly updated this section. (Lines: 129-136)

Comment 4: The MHP includes personalised physiotherapy and exercise, it will be critical to record how this is personalised for future replication, this is not described and may vary considerably. This may need documenting re FITT i.e. frequency, intensity, type and timing so that this can be analysed. This maybe already documented in the current prehabilitation programme but this need clarifying in the protocol as adherence and intensity are important for intervention fidelity and different clinicians can personalise quite differently i.e. are their core exercises or resistance training and what is adapted.

Thank you. We have now provided more information regarding the MHP in the relevant section. (Lines 230-237).